# Prevalence of Post-Concussion-Like Symptoms in the General Injury Population and the Association with Health-Related Quality of Life, Health Care Use, and Return to Work

**DOI:** 10.3390/jcm10040806

**Published:** 2021-02-17

**Authors:** Marjolein van der Vlegel, Suzanne Polinder, Hidde Toet, Martien J.M. Panneman, Juanita A. Haagsma

**Affiliations:** 1Department of Public Health, Erasmus MC, University Medical Center Rotterdam, 3015 GD Rotterdam, The Netherlands; s.polinder@erasmusmc.nl (S.P.); j.haagsma@erasmusmc.nl (J.A.H.); 2Consumer Safety Institute, P.O. Box 75169, 1070 AD Amsterdam, The Netherlands; h.toet@veiligheid.nl (H.T.); m.panneman@veiligheid.nl (M.J.M.P.)

**Keywords:** RPQ, post-concussion symptoms, injury, health-related quality of life, health care utilization, return to work

## Abstract

Little is known about post-concussion-like symptoms in the general injury population and the association of these symptoms with outcome after injury. This study aimed to assess the prevalence of post-concussion-like symptoms in a general injury population and describe the association between post-concussion syndrome (PCS) and health-related quality of life (HRQL), health care use, and return to work. In this longitudinal study of a cohort of injury patients, data were collected 6 and 12 months after their Emergency Department visit. Questionnaires included socio-demographics, health care utilization, return to work and the five-level version of the EuroQol five-dimensional descriptive system (EQ-5D-5L) to measure HRQL. The 12-month questionnaire included the Rivermead Post-Concussion Symptoms Questionnaire (RPQ). In total, 282 (22.0%) of the 1282 patients met the criteria for PCS. Apart from the high prevalence of PCS in patients with head injuries (29.4%), a considerable proportion of non-head injury patients also had PCS (20.6%) a year after injury. Patients with PCS had lower HRQL, lower return to work rates, and higher health care utilization, compared to patients without PCS. This underlines the importance of developing strategies to prevent post-concussion-like symptoms among injury patients, raising awareness among patients and physicians on the occurrence of PCS, early detection of PCS in the general injury population, and development of strategies to optimize recovery in this group of injury patients, ultimately leading to lower the individual and economic burden of injury.

## 1. Introduction

Individuals with a history of a head injury can experience post-concussion symptoms, such as headache, dizziness, and cognitive impairment. Reports on the prevalence of post-concussion symptoms among individuals with a history of mild traumatic brain injury (mTBI) vary widely, from 11% to 82% [1]. The majority of head injury cases (70–90%) are classified mTBI [2]. Most patients recover within a few months, but some patients experience persistent post-concussion symptoms over time [3]. Post-concussion syndrome (PCS) describes the occurrence of these commonly self-reported symptoms that may occur as cognitive complaints (e.g., memory and concentration problems), somatic complaints (e.g., headache, fatigue, dizziness), and emotional and/or behavioral complaints (e.g., depression, anxiety, irritability) [4]. Post-concussion symptoms are a topic of debate since studies showed that the symptoms related to PCS are not specific to head injury as they concluded that post-concussion-like symptoms and PCS are also prevalent in non-head injured patients [5,6,7,8] and in general populations [9,10,11]. Additionally, an overlap exists between post-concussion symptoms and other physical, neurological, and psychiatric conditions, for example, post-traumatic stress disorder (PTSD) [1,5,12]. There are complex interactions between both injury-related and non-injury-related factors, as well as biological, psychological, and social factors, that can lead to post-concussion symptoms [1]. As these symptoms are non-specific to head injury patients, insight in persistent post-concussion symptoms in a general injury population and the impact on health outcomes after injury could optimize recovery and reduce the burden of these symptoms in the entire injury population by raising awareness among patients and physicians on the occurrence of these symptoms. Additionally, a better understanding of patients who are likely to experience post-concussion symptoms after an injury can help in the identification of those patients that need early interventions.

In individuals with a history of mild traumatic brain injury (mTBI), post-concussion symptoms are associated with lower levels of satisfaction of life, lower health-related quality of life (HRQL), and a slower recovery [13,14,15]. The diminished functional outcome of people with PCS and longer recovery period may be associated with an increase in health care utilization, imposing a substantial economic burden on society. However, no previous studies investigated the relationship between post-concussion symptoms and health care utilization. Furthermore, productivity costs may increase the economic burden of PCS, as post-concussion symptoms are associated with reduced return to work in individuals with a history of mTBI [13,16,17]. Although PCS is associated with worse outcomes in mTBI patients, it is unknown whether the same association also exists in the general injury population. This information will improve understanding the impact of post-concussion symptoms on the recovery after injury. Identifying factors associated with a higher prevalence of PCS, and characterizing patient groups at risk of poor outcome, can facilitate in an early detection of post-concussion symptoms and in its turn optimize recovery.

The aims of this study were to: (1) assess the frequency of self-reported post-concussion symptoms and the prevalence of PCS 12 months post-injury in injury patients, (2) assess risk factors for PCS, and (3) compare HRQL, health care utilization and return to work for injury patients with and without PCS.

## 2. Materials and Methods

### 2.1. Study Design and Population

The Dutch Injury Surveillance System (DISS) registers patients with an injury visiting the Emergency Department (ED) of a representative sample of 14 hospitals throughout the Netherlands [18]. A patient follow-up was conducted among a sample of injury patients of all severity levels and who were registered in DISS. Patients were treated at an ED, followed by hospital admission or discharge to home environment. Our study sample was stratified; hospitalized patients and patients with severe and less common injuries were overrepresented. Each selected patient received a questionnaire 6 (T1) and 12 months (T2) after ED visit. This follow-up study was not subject to the Medical Research Involving Human Subjects Act (WMO), as concluded by the Medical Ethics Committee of the Academic Medical Center of Amsterdam (AMC). All participants signed an informed consent form. Patients aged 18 years and older, who had completed all items of the Rivermead Post-Concussion Symptoms Questionnaire (RPQ), the five-level version of the EuroQol five-dimensional descriptive system (EQ-5D-5L) and reported on health care utilization 12 months after injury, were included in this study. Patients were excluded if a proxy (e.g., a family member) filled out the questionnaires.

### 2.2. Measures

Socio-demographic and injury characteristics: Age, sex, type of injury, and injury mechanism were extracted from DISS. The EUROCOST classification scheme identifies 39 injury groups and corresponds to the tenth Revision of the International Classification of Diseases (ICD-10) codes for type of injury [19]. In this study, type of injury was categorized in 14 groups: severe TBI (contusion cerebri, skull fracture) mild TBI (commotion cerebri, trauma capitis), other head injury (head/facial fractures), superficial head injury, spinal cord injury, rib fracture, other thoracic injury, pelvic injury, fracture upper extremity, other injury upper extremity, pelvic injury, hip fracture, fracture of lower extremity, other injury lower extremity, and other injury according to the EUROCOST classification scheme. Severe TBI, mild TBI and other head injuries were categorized as ‘head injury’ and all other injuries were categorized as ‘non-head injuries’. Injury mechanism was categorized as: home- and leisure-related injuries, traffic injuries, sport injuries or occupational injuries. The T1 questionnaire included questions related to: highest attained educational level, living situation and chronic diseases. Living situation was characterized as living alone or living with a spouse/family. Chronic diseases (respiratory diseases, heart disease, previous stroke, diabetes mellitus, hernia, (rheumatoid) arthritis, cancer, other) were categorized into ‘no chronic disease’, ‘1 chronic disease’, ‘2 or more chronic diseases’. Highest attained education was categorized according to the 2011 International Standard Classification of Education (ISCED) into low (primary school, lower secondary school, or lower vocational training); middle (intermediate and higher secondary school, or intermediate vocational training; and high (higher vocational training or university education) educational level [20].

Post-concussion symptoms and PCS: The T2 questionnaire included the Rivermead Post-Concussion Symptoms Questionnaire (RPQ) to identify the existence and severity of post-concussion symptoms. Participants were asked if they, over the last 24 h, suffer from a symptom, in comparison with before the injury. The RPQ relies on self-report of the presence and severity of symptoms and describes 16 post-concussion symptoms, including headaches, dizziness, nausea/vomiting, noise sensitivity, sleep disturbance, fatigue, being irritable, feeling depressed or tearful, feeling frustrated or impatient, forgetfulness, poor concentration, taking longer to think, blurred vision, light sensitivity, double vision, and restlessness. For each symptom, patients could assess the severity of their symptom on a 5-point Likert scale (0 not experienced at all, 1 no more of a problem, 2 mild problem, 3 moderate problem, 4 severe problem). For the total RPQ score, the scores for each symptom are summed, except for scores of 1 (indicating no more of a problem). This results in a total RPQ score ranging from 0 (no symptoms) to 64 (severe symptoms) [21]. There are several classification methods to define PCS [22]. In this study, the criteria described in the ICD-10 were used to classify patients as those having PCS and those not having PCS. Patients had to report at least three of the following symptoms: headaches, dizziness, fatigue, irritability, poor memory, poor concentration, and sleep disturbance. We used a cut-off of ≥2 rating score (mild or higher). Other diagnostic criteria, according to the ICD-10, are reduced tolerance to stress, emotional excitement, or alcohol and a history of TBI, but this information was not available in the RPQ nor other items in the questionnaire and, therefore, is not included as a criteria for PCS in this study. Additionally, it must be emphasized that the RPQ is self-reported and cannot be used to clinically diagnose PCS. Post-concussion syndrome (PCS) has been considered to be present when ≥3 of the core post-concussion symptoms are present. In this study, we also use this definition on patients without a history of head-injury. It is important to note that, to correctly diagnose people with PCS, a clinical examination should take place, and there should be a history of TBI. 

Health-related quality of life: The T1 and T2 questionnaires included the EQ-5D-5L to assess HRQL. The EQ-5D-5L includes items on five dimensions: mobility, self-care, usual activities, pain/discomfort, and anxiety/depression. Each dimension included five severity levels: ‘no problems’, ‘slight problems’, ‘moderate problems’, ‘severe problems’, or ‘extreme problems’ [23]. The patient is asked to indicate their health state in each of the five dimensions. An EQ-5D-5L utility score was calculated using the Dutch EQ-5D-5L value set established from the Dutch population with a score anchored on a scale ranging from 0 (indicating “death”) to 1 (indicating “full health”) [24]. Scores lower than 0 represent states considered to be worse than death. Additionally, a Visual Analogue Scale (EQ-VAS) measured a patient’s self-rated health. The score ranges from 0 (‘worst imaginable health state’) to 100 (‘best imaginable health state’). Furthermore, EQ-5D-5L profiles at 6 months and 12 months were compared using the Paretian classification of Health Change [25]. Respondents were classified as ‘no problems’ (no problems at T1 and T2), ‘no change’ (no change between T1 and T2), ‘improvement’ (improvement in at least one dimension and no worsening in any other dimension), ‘worsening’ (worsening in at least one dimension and no improvement in any other dimension), and ‘mixed change’ (a mix of ‘better’ and ‘worse’ across dimensions).

Health care utilization: Data on hospital admission was extracted from DISS. T1 and T2 questionnaires also included questions on health care use related to: specialist visits, outpatient rehabilitation, general practitioner visits, physiotherapist visits, psychologist visits, and nursing care at home. Patients reported on their health care utilization in the first six months after injury at the 6-month follow-up and on their health care utilization seven to twelve months after injury at the 12-month follow-up.

Working status: Patients who were employed at the time of injury, aged 18–67, reported at T1 and T2 if they were absent from work due to their injury in the last six months, as well as reported if they were still absent at 6 or 12 months post-injury. Return to work was categorized as ‘No absence from work’, ‘return to work within 6 months’, return to work within 7–12 months’, and ‘no return to work at 12 months’.

### 2.3. Statistical Analysis

Aim 1: Descriptive statistics were used to analyze patient and injury characteristics for the total sample, as well as for patients with and without PCS. To compare patients with and without PCS, chi-square tests for categorical variables and student’s t tests for continuous variables were used.

Aim 2: Missing data for education (72 missing), chronic disease (32 missing), and living situation (17 missing) were imputed using multiple imputation with ten datasets using multiple imputations by chained equations (MICE) [26]. Univariate and multivariate logistic regression analysis were used to analyses risk factors for PCS. Age, sex, educational level, living situation, cause of injury, number of injuries, type of injury, chronic diseases, and length of hospital stay were considered risk factors. Factors that were significantly associated with PCS (*p* < 0.05) in the univariate regression models were included in the multivariate model.

Aim 3: Descriptive statistics were used to describe the health-related quality of life, health care utilization and return to work rates for patients with and without PCS. Mann–Whitney U tests and independent sample t-tests were used to evaluate differences in EQ-5D-5L utility and EQ VAS scores and prevalence of problems in each dimension of the EQ-5D-5L between patients with and without PCS. Spearman’s correlation coefficients were used to analyze the correlation between the RPQ and the EQ-5D-5L dimensions and EQ-5D-5L utility and EQ VAS score. To evaluate the differences in EQ-5D-5L utility scores, EQ VAS scores, and health care utilization over time (measured at 6 and 12 months), paired t-test was used. The association between PCS and return to work was assessed using chi-square tests. Statistical significance was determined by a *p*-values of *p* < 0.05.

Pooled results from the imputed datasets were used in the regression model, and complete-case data were used to perform the descriptive analysis. Unweighted and weighted data were presented. The weighted data can be considered representative of an adult injury population who visited the ED’s in the Netherlands. All statistical analysis were performed using SPSS version 25 for Windows (IBM SPSS Statistics, SPSS Inc, Chicago, IL, USA) and R (3.5.3, R Foundation for Statistical Computing, Vienna, Austria, 2019).

## 3. Results

### 3.1. Study Population

In total, 1282 respondents were included in this study (Figure A1). Respondents were slightly younger (61.9 years versus 63.7 years), more often male (46.4% versus 41.9%), had a higher educational level (30.3% versus 23.6% with high educational level), less frequently reported chronic diseases (57.0% versus 51.4% without chronic diseases), and less often lived alone (25.1% versus 33.9%) compared to those lost to follow-up. The characteristics of the study sample are shown in Table 1. The majority of respondents were female (53.6%). The mean age of the respondents was 61.9 (SD 15.6). The most frequently reported cause of injury was a home and leisure accident (54.7%) and head injury was identified in 197 (15.4%) patients. Approximately 40% of patients reported to have one or more chronic diseases.

### 3.2. Prevalence of Post-Concussion-Like Symptoms

The mean total RPQ score was 4.9 (SD: 7.8) (Table 1). The prevalence of PCS in our study population was 22.0%. Of patients with head injury, 28.4% had PCS compared to 20.6% of patients with non-head injuries (*p* = 0.009). Additionally, 40.4% of patients with two or more chronic diseases were classified as having PCS compared to 14.5% for patients without chronic diseases and 28.3% for patients with one chronic disease (*p* < 0.001). Patients with PCS had a significantly lower educational level compared to patients with no PCS (*p* = 0.001). Road traffic accidents were more prevalent among patients who reported PCS (*p* < 0.001). Of patients with severe TBI (*n* = 55), mild TBI (*n* = 145), and other head injuries (*n* = 36), respectively, 38.2%, 24.1%, and 30.6% had PCS (Figure 1). PCS was also common under patients with pelvic injuries (34.8%) and fractures of lower extremity (28.7%). Of patients with non-fracture injuries of the lower extremity, the least number of patients had PCS (14.5%).

Nearly half (48.3%) of patients reported experiencing at least one post-concussion symptom at 1 year post-injury. The most frequently reported symptom by patients with and without head injury was fatigue; respectively, 35.2% and 32.6% of patients had these symptoms (Figure 2). Additionally, taking longer to think (24.2%), poor concentration (23.7%), and forgetfulness (23.7%) were most common in patients with head injury. The most common symptoms for non-head injury patients apart from fatigue were sleep disturbance (23.0%), feeling frustrated or impatient (16.7%), and forgetfulness (16.3%). Patients with head injury showed significantly more feelings of headache, dizziness, noise sensitivity, depression/tearful, poor memory, poor concentration, taking longer to think, light sensitivity, blurred vision, and double vision symptoms compared to patients with non-head injury (*p* < 0.05). For mild TBI patients (*n* = 144), fatigue (32.4%), taking longer to think (23.4%), poor concentration (20.7%), poor memory (20.7%), headache (20.7%), and sleep disturbance (20.0%) were the most common post-concussion-like symptoms. For severe TBI patients, fatigue (47.3%), poor concentration (32.7%), poor memory (32.7%), taking longer to think (30.9%), noise sensitivity (27.3%), and irritability (27.3%) were most frequently reported.

### 3.3. Risk Factors for PCS

Multivariable regression analysis showed that low educational level, road traffic accidents, head injury, higher number of injuries, chronic disease and longer hospital length of stay were significantly associated with PCS (Table 2). Two or more chronic diseases compared to no chronic disease (odds ratio (: 4.1; 95% CI: 2.8; 6.0) and head injury compared to other injuries (odds ratio: 1.5; 95% CI: 1.0; 2.2) were the strongest predictors of PCS in our model. In the subgroup of patients with head injury, having chronic diseases, having no multiple injuries, and longer hospital stay were statistically significant associated with PCS (Table A1 in Appendix A).

### 3.4. Health-Related Quality of Life

The mean EQ-5D-5L utility score for patients with PCS was 0.65 (SD: 0.25) at 6 months post-injury and 0.68 (SD: 0.24) at 12 months post injury, whereas, for patients without PCS, the mean EQ-5D-5L utility scores were, respectively, 0.84 (SD: 0.17) and 0.88 (SD: 0.15) (Table 3). EQ VAS scores were 63.7 (SD: 15.8) and 64.7 (SD: 17.8) for patients with PCS and 77.7 (SD: 15.8) and 79.3 (SD: 16.2) for patients without PCS, respectively, 6 and 12 months post-injury. Both EQ-5D-5L utility score and EQ VAS score at 6 and 12 months differed statistically significantly between patients with and without PCS (*p* < 0.001). Patients with PCS reported significantly more problems on all five EQ-5D-5L dimensions both at 6 months and 12 months after injury (*p* < 0.001). Additionally, for patients without PCS, the prevalence of problems on each dimension decreased significantly at 12 months compared to 6 months after injury. However, the prevalence of problems on all dimensions except ‘usual activities’ did not differ significantly at 6 and 12 months for patients with PCS. For both PCS patients with and without head injury, the majority of patients still experienced problems on the EQ-5D-5L dimensions, 12 months after injury (Figure 3). Furthermore, patients with non-head injuries and PCS more frequently reported problems on the EQ-5D-5L dimensions than patients with head-injuries and PCS, with statistically significant differences in the ‘mobility’ and ‘self-care’ dimensions. At 12 months, the mean EQ-5D-5L utility score for patients with PCS and head injury was 0.72 (SD: 0.21) compared to 0.67 (SD: 0.25) for patients with non-head injury (Table A2). Table 4 shows the distribution of changes according to the Paretian classification of Health change. The proportion of patients that showed overall worsening (22.2%) and mixed change (25.2%) of the EQ-5D-5L profile was higher in patients with PCS compared to patients without PCS (respectively, 13.4% and 9.5%). The unweighted and weighted data showed similar results.

All RPQ items were positively correlated with the EQ-5D-5L dimensions (Figure 4). The strongest correlations were found between the symptom ‘fatigue’ and the ‘daily activities’ dimension (0.555) and between the symptom ‘feeling depressed’ and the ‘anxiety/depression’ dimension (0.519). The total EQ-5D-5L utility score and EQ VAS score were most strongly associated with ‘fatigue’ (respectively, −0.595 and −0.477) and ‘feeling frustrated’ (respectively, −0.507, −0.406). Double vision had the weakest correlation with both the EQ-5D-5L utility score and EQ VAS score.

### 3.5. Health Care Utilization

Table 5 shows the frequency of health care utilization within 12 months after injury for different health care services. Half of all patients were hospitalized (49.1%), and more than half of all patients visited a specialist (64.9%) and/or physiotherapist (59.0%), in the first year after injury. The majority of patients with PCS were hospitalized (56.7%) compared to 47.0% of patients without PCS (*p* < 0.005). The data corrected for stratification showed that 26.7% of patients without PCS 29.4% of patients with PCS were admitted to a hospital. Patients with PCS more frequently made use of all types of health care services compared to patients without PCS (*p* < 0.005). For example, more than half of patients (60.6%) with PCS visited a general practitioner in the first year after injury compared to 32.1% of patients without PCS. In the 7–12 months after injury, 9.8% of patients without PCS had a general practitioner visit compared to 35.5% of patients with PCS (*p* < 0.001). Furthermore, 44.3% of patients with PCS visited a physiotherapist 7–12 months after injury compared to 24.4% of patients without PCS (*p* < 0.001). Patients with PCS and with chronic diseases less often visited a specialist, outpatient rehabilitation, general practitioner physiotherapist, or a psychologist but more often were hospitalized and received nursing care at home compared to patients with PCS and without chronic diseases. These differences were statistically significant for outpatient rehabilitation, psychological care, and nursing care at home (*p* < 0.05).

### 3.6. Return to Work

Figure 5 shows the return to work rates after injury. More than 90% of patients returned to work within 12 months after injury. However, there was a significant difference in return to work rates between patients with and without PCS. Most patients without PCS returned to work within 6 months (84.2%), while only half of patients (50.5%) with PCS returned to work within 6 months (*p* < 0.001). Additionally, nearly a third of patients (31.1%) with PCS were not able to return to work within 12 months after injury, compared to only 4.3% of patients without PCS (*p* < 0.001). There was a moderately strong association between PCS and no RTW within 6 months and 12 months (respectively, φ = 0.325, *p* < 0.001 and φ = 0.360, *p* < 0.001).

## 4. Discussion

This study focused on the prevalence of post-concussion-like symptoms and assessed the association with HRQL, health care utilization and return to work in a general injury population. Overall, post-concussion-like symptoms were common among all injury patients, of which 22.0% had PCS. Apart from the high prevalence of PCS in patients with head injuries (28.4%), also a considerable proportion of non-head injury patients had PCS (20.6%) a year after injury. There are differences in symptom profiles for patients with and without head injury, as somatic problems (e.g., headaches, dizziness) and cognitive problems (e.g., taking longer to think, poor concentration) were more prevalent in head injury patients while emotional and/or behavioral symptoms occurred similarly in head and non-head injury patients. Low educational level, road traffic accidents, head injury, chronic diseases, and hospitalization are associated with PCS and, therefore, are important characteristics for early detection of post-concussion symptoms. Our results showed that patients with PCS had a lower HRQL. Additionally, the RPQ items were significantly correlated with all EQ-5D-5L domains, which also indicates that experiencing post-concussion symptoms is associated with a lower HRQL of the patient. Lastly, we found that patients with PCS more often received health care in the first twelve months after injury and had lower return to work rates compared to patients without PCS.

The prevalence rates of PCS varied with type of injury with lowest rates for non-fracture injury of lower extremity (14.5%) to highest rates for severe TBI (38.2%). The prevalence rate of PCS in patients with head injuries (28.4%) and in patients with non-head injuries (20.6%) were comparable to other studies, for example, the prevalence rates found in a mild TBI versus a non-head population by Lagarde et al. (respectively, 28.7% and 22.9%, based on ≥3 symptoms of RPQ) [5]. Although post-concussion symptoms were more common in patients with head-injury, our results adds to the evidence that post-concussion symptoms are not specific to mild TBI [12]. Interestingly, our results indicate differences in symptom profiles for patients with and without head injury. Each of the cognitive complaints (poor memory, poor concentration, and taking longer to think) and most somatic complaints (headaches, dizziness, blurred vision, double vision, light sensitivity) were significantly more prevalent in head injury patients, whereas most of the emotional complaints (irritably, frustration, restlessness) were similarly present in head and non-head injury patients. Our results suggest that some of the symptoms, mostly somatic and cognitive, are associated more with head injury than emotional/behavioral symptoms, which are generally linked to psychological distress. These findings are similar to the results of a previous study that compared the scores for each RPQ item between patients with a history of mTBI and healthy persons with no history of mTBI. They found that patients with mild TBI had significantly higher scores for headaches, dizziness, nausea, taking long to think, and light sensitivity compared to controls [7]. In another study with chronic pain and mild TBI patients, mild TBI patients had more memory problems, slowed cognition, and light/noise sensitivity than chronic pain patients did [27]. Additionally, fatigue, sleep disturbance, and feeling frustrated were frequently reported symptoms in patients with non-head injuries, which corresponds to the results of a recent study on post-concussion-like symptoms in a general population [11]. For patients with head injury, as well as for the subgroup of patients with a history of mTBI, fatigue was the highest reported symptom, followed by poor memory and poor concentration and taking longer to think. These patterns are in line with previous studies on PCS symptoms among mild TBI patients [22,28,29,30].

Low educational level, road traffic accidents, head injury, chronic diseases, and hospitalization were significantly associated with PCS. This is in line with previous studies [22,31,32]. A systematic review on PCS in mild TBI patients concludes that older age and female sex are significantly associated with the development of PCS [33]. In our study, age was not associated with PCS, but the mean age (61.9 years) in our study was considerably higher compared to previous studies primarily focused on mild TBI patients. Although female sex is associated with PCS in patients with a history of mild TBI, as indicated by previous studies, our results in a general injury population did not find this association. A better understanding of patients who are likely to have PCS after an injury can help in the identification of those patients that need early interventions.

Furthermore, post-concussion-like symptoms in injury patients were associated with worse HRQL, higher health care utilization, and lower return to work rates. Individuals with PCS more frequently reported problems on all EQ-5D-5L dimensions. The EQ-5D-5L utility score of patients without PCS increased significantly between 6 and 12 months after injury, with 12-months EQ-5D-5L utility score comparable with the general Dutch population [24]. However, HRQL scores for patients with PCS did not increase significantly between 6 and 12 months after injury, and they were considerably lower than the EQ-5D-5L utility score of the general Dutch population. Additionally, according to the Paretian classification for profile changes, a higher proportion of patients with PCS reported overall worsening and mixed changes of the EQ-5D-5L profile compared to patients without PCS, suggesting a more complex recovery for these patients. Additionally, all post-concussion-like symptoms and EQ-5D-5L dimensions were correlated, with a similar pattern found in a general population sample by Voormolen et al. [11]. The anxiety/depression and the usual activities dimension had the strongest correlation with fatigue and several emotional post-concussion symptoms, including feeling depressed, irritated, frustrated, and restlessness.

The health care utilization of patients with PCS also indicates that they are not fully recovered, since patients with PCS more often used health care services than patients without PCS use, and more often utilized health care up to 12 months post-injury. Rates of return to work were low for patients with PCS, as only 60% of patients return to work within 1 year after injury. Other studies on patients with a history of TBI also found that patients with PCS had lower return to work rates [34].

This study is unique compared to other studies as we provided information on the prevalence of PCS in the general injury population and additionally showed its impact on HRQL, health care utilization and return to work. This study has several limitations. First, RPQ was self-reported, which could lead to under- or over reporting of symptoms [35]. Second, apart from the RPQ, there was no information available on tolerance to stress, emotional excitement, and alcohol use, which are part of the ICD-10 diagnostic criteria for PCS [36]. Therefore, to diagnose PCS, the RPQ alone cannot be used. This study did not include a pre-injury and 6-month RPQ measurement. It is, therefore, unknown if patients experienced post-concussion symptoms before sustaining their injury. A third limitation of this study was that we were unable to perform longitudinal analysis on PCS since the RPQ was not included in the T1 questionnaire. Lastly, patients included in this study differed significantly from patients lost to follow-up. This could lead to an underestimation of post-concussion-like symptoms since included patients were generally younger, were higher educated, and had less chronic diseases, which were associated with a lower likelihood of PCS.

There is limited evidence for effective interventions targeting post-concussion symptoms (1) but early recognition at the ED of patients with a higher risk for PCS and recognition of post-concussion symptoms could help health care providers in providing optimal care. Patient education on the presence of these symptoms, given as instructions at discharge from the ED, can also facilitate better outcome. A recent study on unfavorable outcomes in patients with mild TBI showed that early multidimensional management involving psychoeducation and cognitive rehabilitation considerably improves the outcome of these patients [37]. Future research could further explore the patterns of cognitive, somatic and emotional symptoms to provide specific interventions to reduce post-concussion symptoms and therewith optimize recovery of injury.

## 5. Conclusions

Post-concussion-like symptoms are prevalent in patients with head and non-head injuries. There are differences in symptom profiles for patients with and without head injury. Most of the emotional complaints were similarly present in head and non-head injury patients, whereas each of the cognitive and most somatic complaints were significantly more prevalent in head injury patients. Compared to patients without PCS, patients with PCS had a lower HRQL, higher health care utilization, and lower return to work rates. This underlines the importance of early detection of post-concussion symptoms in the general injury population and development of strategies to optimize recovery and ultimately lower the individual and economic burden in this group of injury patient.

## Figures and Tables

**Figure 1 jcm-10-00806-f001:**
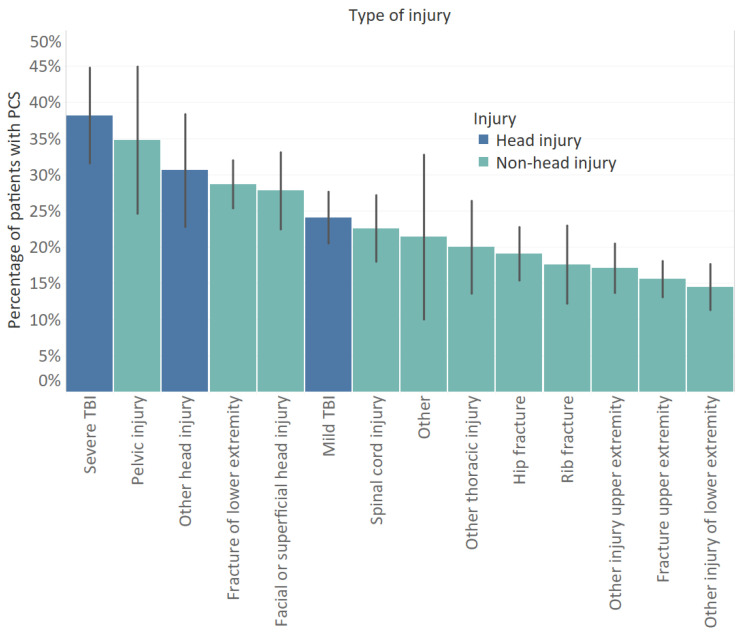
Percentage of patients with post-concussion syndrome (≥3 of the core post-concussion symptoms) for different injury groups. Error bars represent the standard error.

**Figure 2 jcm-10-00806-f002:**
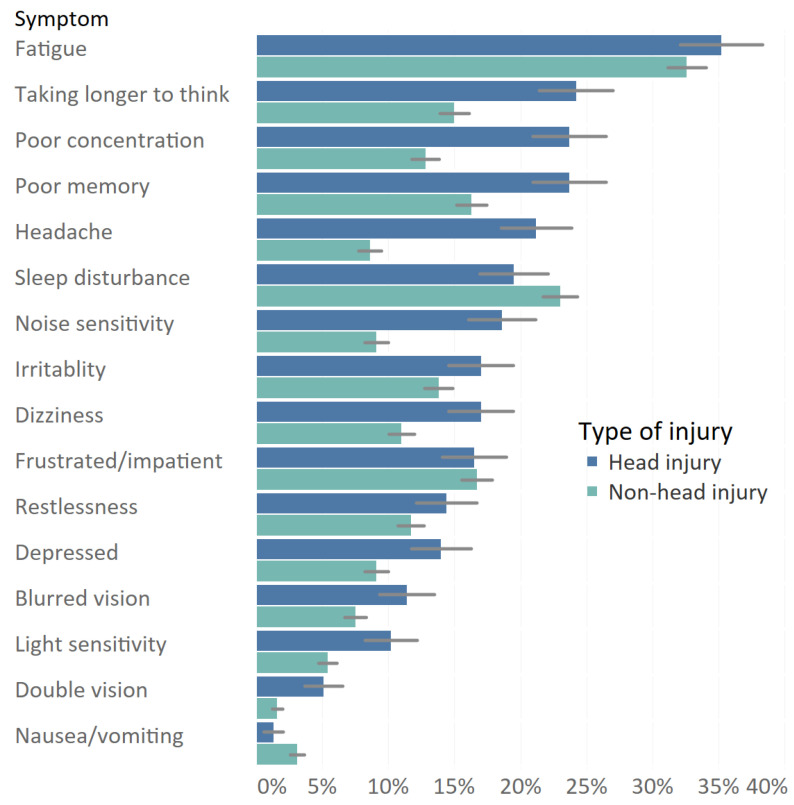
Frequency of post-concussion symptoms at twelve months post-injury. Error bars represent the standard error.

**Figure 3 jcm-10-00806-f003:**
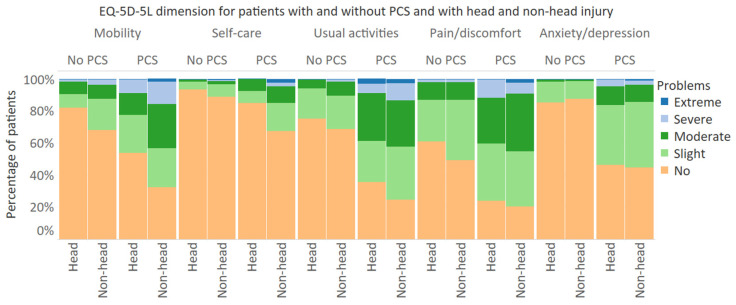
Frequency of responses to EQ-5D-5L of patients with and without head injury, by PCS status. PCS: Post-concussion syndrome (≥3 of the core post-concussion symptoms).

**Figure 4 jcm-10-00806-f004:**
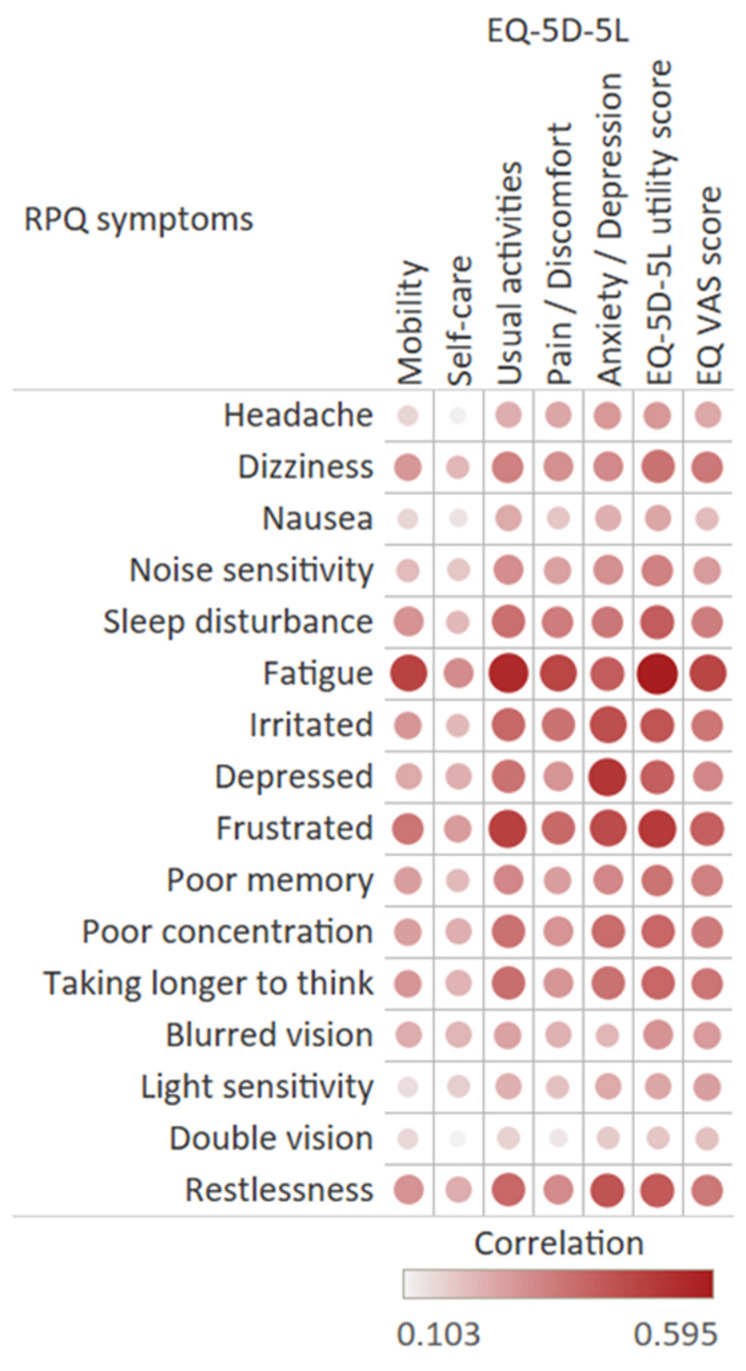
Correlation of RPQ items and EQ-5D-5L dimensions and EQ-5D-5L utility and EQ VAS scores.

**Figure 5 jcm-10-00806-f005:**
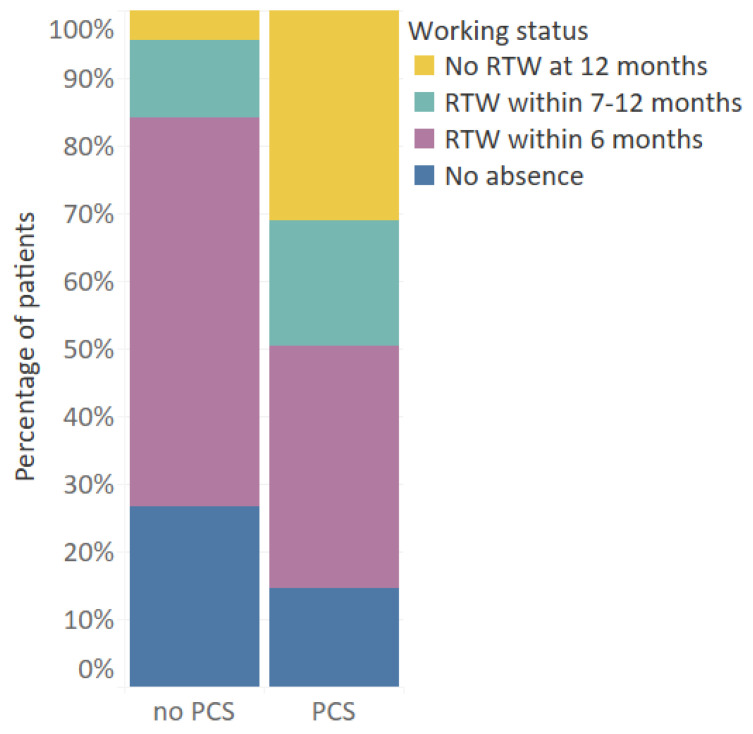
Return to work rates for patients aged 18–67 who had paid employment pre-injury (*n* = 770), by PCS status. PCS: Post-concussion syndrome (≥3 of the core post-concussion symptoms).

**Table 1 jcm-10-00806-t001:** Characteristics of the study population and comparison between patients with and without post-concussion syndrome (≥3 of the core post-concussion symptoms).

Characteristic	Group	Total (*n* = 1282)	No PCS (*n* = 1000)	PCS (*n* = 282)	Respondents (Weighted) ^a^
Age in years, mean (SD)		61.9 (15.7)	61.5 (15.6)	63.1 (16.1)	60.9 (15.2)
Sex, *n*	Male	595 (46.4%)	476 (47.6%)	119 (42.2%)	44.1%
	Female	687 (53.6%)	524 (52.4%)	163 (57.8%)	55.9%
Educational level, *n* ^b^	Low	506 (39.5%)	371 (37.1%)	135 (47.9%)	38.7%
	Middle	338 (26.4%)	270 (27.0%)	68 (24.1%)	31.8%
	High	367 (28.6%)	306 (30.6%)	61 (21.6%)	29.5%
Living situation, *n* ^c^	Alone	317 (24.7%)	239 (23.9%)	78 (27.7%)	29.3%
	Not alone	948 (73.9%)	748 (74.8%)	200 (70.9%)	70.7%
Injury mechanism, *n*	Home and leisure accident	700 (54.6%)	553 (55.3%)	147 (52.1%)	58.6%
	Road traffic accident	350 (27.3%)	253 (25.3%)	97 (34.4%)	23.7%
	Sports accident	155 (12.1%)	137 (13.7%)	18 (6.4%)	10.6%
	Occupational accident	77 (6.0%)	57 (5.7%)	20 (7.1%)	7.1%
Type of injury, *n*	Head injury	236 (18.4%)	169 (16.9%)	67 (23.8%)	13.0%
	Other injuries	1046 (81.6%)	831 (83.1%)	224 (76.2%)	87.0%
Chronic disease, *n* ^d^	No chronic disease	712 (55.5%)	609 (60.9%)	103 (36.5%)	54.5%
	1 chronic disease	350 (27.3%)	251 (25.1%)	99 (35.1%)	25.5%
	2 or more chronic diseases	188 (14.7%)	112 (11.2%)	76 (27.0%)	20.0%
Post-concussion syndrome (PCS)	Yes	282 (22.0%)			19.3%
	No	1000 (78.0%)			80.7%
RPQ total score					
mean (SD)		4.9 (7.8)	1.5 (2.7)	17.0 (8.2)	4.5 (7.6)
median (IQR)		0 (0–7)	0 (0–2)	15 (11–21)	0 (0–6)

SD: Standard Deviation, IQR: Inter quartile range, RPQ: Rivermead Post-Concussion Symptoms Questionnaire, PCS: Post-concussion syndrome (≥3 of the core post-concussion symptoms). ^a^ Data are corrected for stratification, and are representative of an adult population of injured patients who visited an emergency department in The Netherlands. ^b^ 71 missing values (5.5%). ^c^ 17 missing values (1.3%). ^d^ 32 missing values (2.5%).

**Table 2 jcm-10-00806-t002:** Characteristics of the study population and comparison between patients with and without post-concussion syndrome (≥3 of the core post-concussion symptoms).

Characteristic	Group	Univariate		Multivariate	
		Odds ratio	95% CI	Odds ratio	95% CI
Age		1.007	0.998; 1.016		
Sex	Male	Ref			
	Female	1.244	0.953; 1.625		
Living situation	Not Alone	Ref			
	Alone	1.219	0.903; 1.644		
Educational level	Low	Ref		Ref	
	Middle	0.679 *	0.490; 0.940	0.807	0.573; 1.136
	High	0.552 *	0.391; 0.778	0.664 *	0.462; 0.954
Injury mechanism	Home and leisure accident	Ref		Ref	
	Road traffic accident	1.442 *	1.072; 1.940	1.493 *	1.079; 2.066
	Sports accident	0.494 *	0.293; 0.835	0.747	0.430; 1.300
	Occupational accident	1.320	0.769; 2.267	1.570	0.884; 2.789
Number of injuries	1	Ref			
	2	1.210	0.865; 1.692	1.013	0.703; 1.460
	≥3	1.555 *	1.001; 2.415	1.174	0.707; 1.948
Type of injury	Other injuries	Ref		Ref	
	Head injury	1.532 *	1.112; 2.111	1.485 *	1.023; 2.155
Chronic disease	No chronic disease	Ref		Ref	
	1 chronic disease	2.337 *	1.710; 3.194	2.403 *	1.739; 3.319
	2 or more chronic diseases	3.932 *	2.740; 5.642	4.126 *	2.825; 6.026
Length of hospital stay		1.066 *	1.037; 1.095	1.058	1.029; 1.087

CI: Confidence Interval; * statistically significant.

**Table 3 jcm-10-00806-t003:** Mean T1 (6 months) and T2 (12 months) five-level, five-dimensional descriptive system (EQ-5D-5L) utility and visual analogue scale (EQ VAS) scores and percentage of patients reporting problems on EQ-5D-5L dimensions, for patients with and without post-concussion syndrome (≥3 of the core post-concussion symptoms).

	6 Months after Injury		12 Months after Injury	
Health-Related Quality of Life	PCS	No PCS	PCS	No PCS
Mean (SD) EQ-5D-5L utility score	0.65 (0.25)	0.84 (0.17)	0.68 (0.24)	0.88 (0.15)
Problems with mobility	64.1%	39.2%	62.4%	29.6%
Problems with self-care	32.5%	15.5%	28.4%	10.4%
Problems with usual activities	82.1%	44.0%	72.7%	30.2%
Pain/discomfort	82.4%	63.1%	78.7%	48.8%
Anxiety/depression	52.5%	15.8%	55.0%	12.9%
Mean (SD) EQ VAS score	63.7 (17.4)	77.7 (15.8)	64.7 (17.4)	79.2 (16.2)
Weighted data ^a^				
Mean (SD) EQ-5D-5L utility score	0.67 (0.24)	0.84 (0.17)	0.70 (0.23)	0.88 (0.23)
Problems with mobility	59.2%	36.8%	60.2%	29.5%
Problems with self-care	29.1%	18.7%	22.5%	14.8%
Problems with usual activities	80.3%	42.2%	74.6%	30.6%
Pain/discomfort	84.2%	63.3%	77.7%	46.4%
Anxiety/depression	50.7%	13.5%	56.8%	16.4%
Mean (SD) EQ VAS score	65.9 (16.9)	77.9 (15.7)	66.3 (16.8)	80.0 (16.8)

SD: standard deviation, PCS: Post-concussion syndrome (≥3 of the core post-concussion symptoms). ^a^ Data are corrected for stratification, and are representative of an adult population of injured patients who visited an emergency department in The Netherlands.

**Table 4 jcm-10-00806-t004:** Health profile change over time according to Paretian classification for profiles changes, for total population and patients with and without post-concussion syndrome (PCS).

Paretian Classification of Health Change	Total	No PCS	PCS
No problems	20.7%	24.7%	2.6%
No change	8.9%	10.5%	7.0%
Improve	42.1%	41.9%	43.0%
Worsen	15.4%	13.4%	22.2%
Mixed change	12.9%	9.5%	25.2%
Weighted data ^a^			
No problems	23.5%	28.5%	2.3%
No change	9.2%	10.3%	4.6%
Improve	38.4%	37.3%	43.2%
Worsen	12.8%	10.3%	23.3%
Mixed change	16.1%	13.6%	26.6%

PCS: Post-concussion syndrome (≥3 of the core post-concussion symptoms). ^a^ Data are corrected for stratification, and are representative of an adult population of injured patients who visited an emergency department in The Netherlands.

**Table 5 jcm-10-00806-t005:** Health care use within twelve months after injury by patients with and without post-concussion syndrome (PCS), by chronic disease status.

Service	Total Sample			No Chronic Disease		With Chronic Disease	
	No PCS	PCS	No PCS		PCS	No PCS	PCS
Hospitalization	47.0%	56.7%	45.2%		53.4%	48.5%	58.3%
Post-discharge health care utilization							
Specialist	62.4%	73.8%	61.2%		76.7%	63.4%	72.6%
Outpatient rehabilitation	7.3%	18.4%	6.6%		27.2%	9.1%	13.7%
General practitioner	32.1%	60.6%	30.9%		63.1%	34.7%	60.0%
Physiotherapist	56.9%	66.3%	57.0%		69.9%	56.6%	64.6%
Psychologist	4.5%	20.2%	4.4%		28.2%	5.0%	16.0%
Nursing care at home	15.2%	30.5%	10.7%		23.3%	22.6%	35.4%
Weighted data ^a^							
Hospitalization	26.7%	29.4%	20.2%		24.8%	35.0%	31.1%
Post-discharge health care utilization							
Specialist	57.9%	71.5%	62.0%		78.3%	51.4%	67.2%
Outpatient rehabilitation	5.2%	15.9%	5.9%		25.1%	4.5%	9.5%
General practitioner	29.5%	62.0%	27.7%		66.1%	31.8%	59.9%
Physiotherapist	46.7%	59.5%	51.4%		66.1%	40.0%	54.9%
Psychologist	4.6%	19.6%	3.6%		26.9%	6.2%	14.6%
Nursing care at home	9.5%	23.7%	6.8%		16.6%	12.6%	29.3%

PCS: Post-concussion syndrome (≥3 of the core post-concussion symptoms). ^a^ Data are corrected for stratification, and are representative of an adult population of injured patients who visited an emergency department in The Netherlands.

## Data Availability

The data that support the findings of this study are only available on request. Requests may be sent to: Martien Panneman (m.panneman@veiligheid.nl). The data presented in this study are not publicly available due to information that could compromise the privacy of research participants.

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
