# Peer review of "Prevalence of Post-Concussion-Like Symptoms in the General Injury Population and the Association with Health-Related Quality of Life, Health Care Use, and Return to Work"

_jcm, 2021, doi:10.3390/jcm10040806_

Round 1

Reviewer 1 Report

This paper is an epidemiological study of the prevalence, risk factors, and correlates of PCS in injured patients. It has many strengths such as the inclusion of important markers of recovery such as health-related quality of life, healthcare utilization, and return to work status. However, several limitations temper my enthusiasm including the lack of clear definition of TBI severity, lack of inclusion of mild versus severe head injury comparisons, and inclusion of those with multiple injuries in both groups. Also, the statistical analysis and results section would benefit from greater organization and more clear delineations as to which analyses are addressing which aim and how the regression models were run.

Introduction

-The authors state that the majority of patients with head injury experience post-concussion symptoms but cite a manuscript stating, “Prevalence rates of post-concussion symptoms vary between 11 and 82%” suggesting that may be an overstatement. A more specific estimate is warranted as well as a short discussion on the nuances following different severities of TBI.

-Using the terms “individuals/patients with a history of traumatic brain injury” is preferred to “TBI population” or “TBI patients” because TBI is an event not a condition.

-The authors should specify the severity of TBI when describing studies throughout the introduction. For example, it seems like the majority of studies they are describing involve individuals with history of mild TBI.

Materials & Methods

-Please define the abbreviation, “WMO”.

-RPQ and EQ-5D-5L should be defined before an abbreviation is used.

-Is there a full name for the “EURO-COST”? If so, please spell it out.

-More information would be useful regarding how TBIs were categorized into mild and severe (e.g., duration of LOC, duration of PTA, GCS).

-Include that the RPQ specifically asks about change in symptoms since the injury, rather than just severity of these problems.

-ICD-10 criteria for PCS also includes a history of TBI, so that should be included in other criteria that were not used to categorize PCS because those without history of TBI were included in the study.

-More information  on the EQ-5D-5L would be helpful as readers may not be familiar with this measure (e.g., number of items, psychometric properties, sample items for each dimension, whether higher or lower scores indicate better or worse quality of life)

-Were “employed patients” employed at the time of injury? If so, please specify that.

-This section would benefit from greater organization regarding which measures were given at T1 versus T2- maybe a table would be helpful?

-The statistical analysis section is also a bit confusing and could benefit from including which analyses correspond to which aim laid out in the introduction.

Results

-Further description of what is meant by “other head injury” would be useful. Additionally, it would be useful to know whether there were significant differences in PCS  and specific symptoms between the different severities of TBI as mild and severe TBIs differ significantly in terms of their expected recovery time. Also, how many had mild vs. severe TBIs?

-It would be helpful to have a breakdown of the most common postconcussive symptoms by mild and severe as well.

-Sentence on page 7 starting on line 226 needs to be reworked for clarity.

-It is unclear how regression models were run and what “univariable” versus “multivariable” is referring to in the table. Here too it would be helpful to have the different severities of TBI uniquely put in as predictors, although sample size will likely be small.

Discussion

-Because the RPQ was given at the 12 month time point and QOL was given at the 6 and 12 month time points, it seems like a stretch to say that experiencing postconcussive symptoms decreases the HRQL of the patient.

-Without listing which symptoms were most common in mild versus severe TBI, it seems difficult to draw comparisons between cited literature suggesting that patient with history of mild TBI had significantly higher scores on somatic/cognitive symptoms.

-Do discrepant findings with regard to sex have to do with not breaking TBI up by severity in this study?

-There should be a citation included for statement made on page 13 starting on line 395.

-The fact that individuals with history of head injury were included in the non head injury group undermines the results of the study. Replication of the analyses excluding those with multiple injuries to ensure that findings remain significant would lend more confidence to the results.

-There is some evidence to suggest that early psychoeducation regarding expected recovery following mild TBI improves outcomes- a brief review of this literature in the discussion seems warranted given that the authors touch on clinical implications of their findings.

-Do those excluded for no RPQ/HRQOL/healthcare differ from those who were not excluded?

Tables

-In Table 1, PCS should be defined in the notes section below the table because it is used in two places but only defined in one

-It looks like Table 2 is labeled Table 1

-There are two Table 4s

Figures

-There are two Figure 1s

Author Response

Response to reviewer 1 comments

  1. This paper is an epidemiological study of the prevalence, risk factors, and correlates of PCS in injured patients. It has many strengths such as the inclusion of important markers of recovery such as health-related quality of life, healthcare utilization, and return to work status. However, several limitations temper my enthusiasm including the lack of clear definition of TBI severity, lack of inclusion of mild versus severe head injury comparisons, and inclusion of those with multiple injuries in both groups. Also, the statistical analysis and results section would benefit from greater organization and more clear delineations as to which analyses are addressing which aim and how the regression models were run.

Response 1.  We thank the author for these thoughtful comments.  Below you find a point-by-point response.

Introduction

  1. The authors state that the majority of patients with head injury experience post-concussion symptoms but cite a manuscript stating, “Prevalence rates of post-concussion symptoms vary between 11 and 82%” suggesting that may be an overstatement. A more specific estimate is warranted as well as a short discussion on the nuances following different severities of TBI.

Response 2.  We have rewritten the first sentences of the introduction and included a sentence on (line 31): “Individuals with a history of a head injury can experience post-concussion symptoms, such as headache, dizziness and cognitive impairment. Reports on the prevalence of post-concussion symptoms among individuals with a history of mild traumatic brain injury (mTBI) vary widely, from 11% to 82%.(1). The majority of head injury cases (70-90%) are classified mTBI.”

  1. Using the terms “individuals/patients with a history of traumatic brain injury” is preferred to “TBI population” or “TBI patients” because TBI is an event not a condition.

Response 3.  Throughout the article we included the use of ‘patients with a history of TBI’ instead of ‘TBI patients/population’.
See line 55, 63, 401, 445.

  1. The authors should specify the severity of TBI when describing studies throughout the introduction. For example, it seems like the majority of studies they are describing involve individuals with history of mild 

Response 4.  Previous studies mainly focused on mild TBI. We therefore specified this in the introduction at line 34, line 55, line 63 and line 64.

Materials & Methods

  1. Please define the abbreviation, “WMO”.

Response 5.  To clarify, we changed the sentence to: “This follow-up study was not subject to the Medical Research Involving Human Subjects Act (WMO), as concluded by the Medical Ethics Committee of the Academic Medical Center of Amsterdam (AMC).” (line 83)

  1. RPQ and EQ-5D-5L should be defined before an abbreviation is used.

Response 6.  We defined the abbreviations at line 87 (Rivermead Post-Concussion Symptoms Questionnaire (RPQ), the 5-level EQ-5D version (EQ-5D-5L))

  1. Is there a full name for the “EURO-COST”? If so, please spell it out.

Response 7.  EUROCOST is the name of the classification scheme to identify injury groups. It is not an abbreviation.

  1. More information would be useful regarding how TBIs were categorized into mild and severe (e.g., duration of LOC, duration of PTA, GCS).

Response 8.  We included additional information in the methods on which injuries were categorized as mild and severe. “The EUROCOST classification scheme identifies 39 injury groups and corresponds to the ICD-10 codes for type of injury (18). In this study type of injury was categorized in 14 groups: severe TBI (contusion cerebri, skull fracture) , mild TBI (commotion cerebri, trauma capitis), other head injury (head/facial fractures), superficial head injury, spinal cord injury, rib fracture, other thoracic injury, pelvic injury, fracture upper extremity, other injury upper extremity, pelvic injury, hip fracture, fracture of lower extremity, other injury lower extremity and other injury according to the EUROCOST classification scheme.” (line 93).

  1. Include that the RPQ specifically asks about change in symptoms since the injury, rather than just severity of these problems.

Response 9.  We included the following sentence (line 116): “Participants were asked if they, over the last 24 hours, suffer from a symptom, in comparison with before the injury.”

  1. ICD-10 criteria for PCS also includes a history of TBI, so that should be included in other criteria that were not used to categorize PCS because those without history of TBI were included in the study.

Response 10.  To emphasize that the criteria used to categorize PCS in our study are not all ICD-10 criteria, we included the following sentences to the method section: “Other diagnostic criteria according to the ICD-10 are are reduced tolerance to stress, emotional excitement, or alcohol and a history of TBI, but this information was not available in the RPQ or other items in the questionnaire and therefore not included as a criteria for PCS in this study. Additionally, it must be emphasized that that the RPQ is self-reported and cannot be used to clinically diagnose PCS. “ (line 132)

  1. More information  on the EQ-5D-5L would be helpful as readers may not be familiar with this measure (e.g., number of items, psychometric properties, sample items for each dimension, whether higher or lower scores indicate better or worse quality of life)

Response 11.  We added extra information on the EQ-5D-5L in the method section: “ The patient is asked to indicate their health state in each of the five dimensions. An EQ-5D-5L utility score was calculated using the Dutch EQ-5D-5L value set established from the Dutch population with a score anchored on a scale ranging from 0 (indicating “death”) to 1 (indicating “full health”) (23). Scores lower than 0 represent states considered to be worse than death.” (line 143)

  1. Were “employed patients” employed at the time of injury? If so, please specify that.

Response 12.  Indeed, this includes patients who were employed prior to the injury. We specified this: “Patients who were employed at the time of injury,” (line 166)

  1. This section would benefit from greater organization regarding which measures were given at T1 versus T2- maybe a table would be helpful?

Response 13.  For each measure, we clearly stated if the measure was included in the T1 ( 6 months ) and/or T2 (12 months) questionnaire. (line 104, line 159, line 167)

  1. The statistical analysis section is also a bit confusing and could benefit from including which analyses correspond to which aim laid out in the introduction.

Response 14.  We re-organized the statistical analysis section and stated which analyses corresponded to which aim (aim 1: line 173, aim 2: line 177, aim 3: line 186)

Results

  1. Further description of what is meant by “other head injury” would be useful. Additionally, it would be useful to know whether there were significant differences in PCS  and specific symptoms between the different severities of TBI as mild and severe TBIs differ significantly in terms of their expected recovery time. Also, how many had mild vs. severe TBIs? It would be helpful to have a breakdown of the most common postconcussive symptoms by mild and severe as well.

Response 15. We included additional information on the types of injury in the methods section. “severe TBI (contusion cerebri, skull fracture) , mild TBI (commotion cerebri, trauma capitis), other head injury (head/facial fractures)” (line 94). Other head injuries included fractures on the head/face for example jaw fractures. In the results section we included the number of patients with TBI: Of patients with severe TBI (n=55), mild TBI (n=145) and other head injuries (n=36) respectively 38.2%, 24.1% and 30.6% had PCS (Figure 1) (line 232). To provide some additional information on symptoms by TBI severity we include the following in the results section:

For mild TBI patients (n=144), fatigue (32.4%), taking longer to think (23.4%), poor concentration (20.7%), poor memory (20.7%), headache (20.7%) and sleep disturbance (20.0%) were the most common post-concussion-like symptoms. For severe TBI patients, fatigue (47.3%), poor concentration (32.7%), poor memory (32.7%), taking longer to think (30.9%), noise sensitivity (27.3%) and irritability (27.3%) were most frequently reported (line 253).

  1. Sentence on page 7 starting on line 226 needs to be reworked for clarity.

Response 16. We have rewritten this sentence to: “In the subgroup of patients with head injury, having chronic diseases, having no multiple injuries and longer hospital stay were statistically significant associated with PCS (Table A1).”

  1. It is unclear how regression models were run and what “univariable” versus “multivariable” is referring to in the table. Here too it would be helpful to have the different severities of TBI uniquely put in as predictors, although sample size will likely be small.

Response 17. To clarify, we included the following sentences in the method section: “Univariate and multivariate logistic regression analysis were used to analyses risk factors for PCS. Age, sex, educational level, living situation, cause of injury, type of injury, chronic diseases and length of hospital stay were considered risk factors. Factors that were significantly associated with PCS (p<0.05) in the univariate regression models were included in the multivariate model. (line 177)

Discussion

  1. Because the RPQ was given at the 12 month time point and QOL was given at the 6 and 12 month time points, it seems like a stretch to say that experiencing postconcussive symptoms decreases the HRQL of the patient.

Response 18. We agree with the reviewer that we cannot state with certainty that experiencing post concussive symptoms decreases the HRQL of the patients based on our results. We therefore slightly changed the following sentence to: “Additionally, the RPQ items were significantly correlated with all EQ-5D-5L domains, which also indicates that experiencing post-concussion symptoms is associated with a lower HRQL of the patient”. (line 380)

  1. Without listing which symptoms were most common in mild versus severe TBI, it seems difficult to draw comparisons between cited literature suggesting that patient with history of mild TBI had significantly higher scores on somatic/cognitive symptoms.

Response 19. We listed the most common symptoms in the results section (line 253) and we included additional information in the discussion section for a better comparison with the literature. Line 409: “For patients with head injury as well as for the subgroup of patients with a history of mTBI, fatigue was the highest reported symptom, followed by poor memory and poor concentration and taking longer to think. These patterns are in line with previous studies on PCS symptoms among mild TBI patients (21, 27-29).”

  1. Do discrepant findings with regard to sex have to do with not breaking TBI up by severity in this study?

Response 20. Previous studies have indeed shown that female sex is associated with PCS in patients with a history of mTBI. Because our study did not specifically focus on mTBI patients but on the general injury we should be cautious with the comparison with these previous studies. To specify we included the following sentence in the discussion section: “Although female sex is associated with PCS in patients with a history of mild TBI as indicated by previous studies, our results in a general injury population did not find this association.” (line 419)

  1. There should be a citation included for statement made on page 13 starting on line 395.

Thank you for pointing this out. We included a citation at line 446.

  1. The fact that individuals with history of head injury were included in the non head injury group undermines the results of the study. Replication of the analyses excluding those with multiple injuries to ensure that findings remain significant would lend more confidence to the results.

Response 22. We thank the author for this important note. In case of multiple injuries, a hierarchy was used to determine the most severe injuries. To make the distinction between the head-injury group and non-head injury group more clear we redid the analysis including all head-injured patients (also if another injury was prioritized) against those without any head-injury diagnoses. In the new analysis, individuals with a history of head injury were all in the head injury category. Figure 1, figure 2, table 2 with the regression analysis and figure 3 and table A1 and A2 were changed .

  1. There is some evidence to suggest that early psychoeducation regarding expected recovery following mild TBI improves outcomes- a brief review of this literature in the discussion seems warranted given that the authors touch on clinical implications of their findings.

We included the following sentence on early psychoeducation: “A recent study on unfavorable outcomes in patients with mild TBI showed that early multidimensional management involving psychoeducation and cognitive rehabilitation considerably improves the outcome of these patients (37)”. ( line 470).

  1. Do those excluded for no RPQ/HRQOL/healthcare differ from those who were not excluded?

We included extra information on the comparison of respondents to those lost to follow up. In the results section we added the following: line 207: “). Respondents were slightly younger (61.9 years versus 63.7 years), more often male (46.4% versus 41.9%), had a higher educational level (30.3% versus 23.6% with high educational level), less frequently reported chronic diseases (57.0% versus 51.4% without chronic dis-eases) and less often lived alone (25.1% versus 33.9%) compared to those lost to follow-up.”

In the discussion we added the following limitation: “Lastly, patients included in this study differed significantly from patients lost to follow-up. This could lead to an underestimation of post-concussion symptoms since included patients were generally younger, higher educated and with less chronic diseases, which were associated with a lower likelihood of PCS” (line 457)

Tables and figures

  1. In Table 1, PCS should be defined in the notes section below the table because it is used in two places but only defined in one

Response 25. We defined PCS in the notes of table 1 at line 219.

  1. It looks like Table 2 is labeled Table 1, There are two Table 4s, There are two Figure 1s

Response 26. Thank you for these corrections. Table 1 is change to table 2 at line 274. Table 4 is changed to table 5 at line 348. Figure 1 is changed to figure 2 at line 261.

Reviewer 2 Report

The authors have compiled a significant amount of data supporting the claim that patients with post-concussive symptoms have worse outcomes in terms of quality of life, return to work, and health care utilization during the year of follow-up. The authors have included many relevant graphs which add to the presentation of this data impressively. Indeed, there is a lot of work showing the deleterious effects of PCS. The text is very readable and flows well.

However, the conclusions concerning which patients are at risk for PCS is not necessarily supported by the data. In particular, I am not sure how patients with only extremity injuries could manifest PCS. They may indeed have fatigue or other non-specific symptoms which fall under the umbrella of PCS, but it is incorrect to define them as post-concussive if no concussion existed. The authors do mention that their methods include utilizing only the most severe injury diagnosis and that their cohort had 25% multitrauma patients. Could these extremity-injured patients have fallen into this category? Could an analysis be done looking at all head-injured patients against those without any head-injury diagnoses, or does the registry not allow for that? I think that this would make the conclusions significantly more powerful rather than what has been presented here. Additionally, data on length of stay for admitted patients would likely support this as well and may serve as another predictive variable to discern who might have PCS and who might not.

All in all, this is a study which presents some very interesting conclusions, but there should be an attempt to clean up the two cohorts. 

Author Response

Response to reviewer 2 comments

  1. The authors have compiled a significant amount of data supporting the claim that patients with post-concussive symptoms have worse outcomes in terms of quality of life, return to work, and health care utilization during the year of follow-up. The authors have included many relevant graphs which add to the presentation of this data impressively. Indeed, there is a lot of work showing the deleterious effects of PCS. The text is very readable and flows well. However, the conclusions concerning which patients are at risk for PCS is not necessarily supported by the data. In particular, I am not sure how patients with only extremity injuries could manifest PCS. They may indeed have fatigue or other non-specific symptoms which fall under the umbrella of PCS, but it is incorrect to define them as post-concussive if no concussion existed.

Response 1. We thank the reviewer for their useful comments. We agree with the reviewer that a patient without a history of TBI cannot manifest PCS. Because we look at post-concussion symptoms in a population including patients without a history of TBI, we used the term ‘post-concussion-like symptoms’ instead of ‘post-concussion symptoms’ when talking about the symptoms that occur in patients without a history of TBI. (line 2 (title), line 42, 225, 255, 267, 269, 408, 425, 436,  458, 477)

To emphasize that the criteria used to categorize PCS in our study are not all ICD-10 criteria, we included the following sentences to the method section: “Other diagnostic criteria according to the ICD-10 are intolerance of stress, emotion or alcohol and a history of TBI, but this information was not available in the RPQ or other items in the questionnaire and therefore not included as a criteria for PCS in this study. Additionally, it must be emphasized that that the RPQ is self-reported and cannot be used to clinically diagnose PCS. “ (line 132).

  1. The authors do mention that their methods include utilizing only the most severe injury diagnosis and that their cohort had 25% multitrauma patients. Could these extremity-injured patients have fallen into this category? Could an analysis be done looking at all head-injured patients against those without any head-injury diagnoses, or does the registry not allow for that? I think that this would make the conclusions significantly more powerful rather than what has been presented here.

Response 2. We thank the author for this important note and our registry does allow for the suggestion of the reviewer. Of the patients with multitrauma in the non-head category, 39 (16.0%) patients also had an head injury. To make the distinction between the head-injury group and non-head injury group more clear we redid the analysis including all head-injured patients (also if another injury was prioritized when a patient had multiple injuries) against those without any head-injury diagnoses. In the new analysis, individuals with a history of head injury were all in the head injury category.

Figure 1, figure 2, table 2 with the regression analysis and figure 3 were changed.

  1. Additionally, data on length of stay for admitted patients would likely support this as well and may serve as another predictive variable to discern who might have PCS and who might not. All in all, this is a study which presents some very interesting conclusions, but there should be an attempt to clean up the two cohorts. 

Response 3. We agree with the suggestion of the reviewer to include length of stay as a predictive variable. We repeated the logistic regression analysis and included length of hospital stay and number of injuries as additional predictive factors (line 274)

Round 2

Reviewer 2 Report

This manuscript has certainly been improved with the addition of the new logistic regression analysis including LOS and number of injuries. Unfortunately, the use of the phrase concussion-like symptoms seems to be almost arbitrary and synonymous with PCS. For example, in 3.1, it the term PCS appears be mean the same thing. Again, patients never having a concussion are labeled with having post-concussion syndrome. If this is the correct parlance, perhaps something of an explanation can be made in the introduction or methods indicating why this is the correct term. At least it still seems to this reader that it is confusing. 

Author Response

9-2-2021

Response to reviewer 2

Reviewer 2 comments: This manuscript has certainly been improved with the addition of the new logistic regression analysis including LOS and number of injuries. Unfortunately, the use of the phrase concussion-like symptoms seems to be almost arbitrary and synonymous with PCS. For example, in 3.1, it the term PCS appears be mean the same thing. Again, patients never having a concussion are labeled with having post-concussion syndrome. If this is the correct parlance, perhaps something of an explanation can be made in the introduction or methods indicating why this is the correct term. At least it still seems to this reader that it is confusing.

Answer:

We thank the reviewer for pointing out that additional explanation is needed on the use of the term PCS. We agree that non-head injured people cannot by definition have PCS. We therefore used the phrases post-concussion-like symptoms and post-concussion symptoms in the manuscript in places discussing the separate symptoms. However, we also included a comparison of patients with ≥3 ICD-10 post-concussion symptoms (with a severity ≥2) to patients with no or fewer post-concussion symptoms based on baseline characteristics, health-related quality of life, health care utilization and return to work. For these comparisons, we categorized patients in two groups, using the terms no PCS and PCS, for analysis purposes only. We agree that it should be clear that general injury patients cannot be clinically diagnosed with PCS. We define PCS in this study as:  ‘having ≥3 of the core post-concussion symptoms’

For further clarification we included the following in the method section line 136:

“Post-concussion syndrome (PCS) has been considered to be present when 3 of the core post-concussion symptoms are present. In this study we use this definition also on patients without a history of head-injury. It is important to note that to correctly diagnose people with PCS, a clinical examination should take place and there should be a history of TBI.”

In each table in the results section where the abbreviation PCS is used we added as a footnote (line 223, line 280, line 312, line 320, line 324, line 360, line 376) : PCS=post-concussion syndrome ( 3 of the core post-concussion symptoms).

Other studies on post-concussion-like symptoms in non-head injury patients or the general population also use the phrase post-concussion-like symptoms and PCS (1, 2).

  1. Voormolen DC, Cnossen MC, Polinder S, Gravesteijn BY, Von Steinbuechel N, Real RGL, Haagsma JA. Prevalence of post-concussion-like symptoms in the general population in Italy, The Netherlands and the United Kingdom. Brain injury. 2019;33(8):1078-86.
  2. Zakzanis KK, Yeung E. Base rates of post-concussive symptoms in a nonconcussed multicultural sample. Archives of clinical neuropsychology. 2011;26(5):461-5.